# UK consultants' experiences of the decision-making process around referral to intensive care: an interview study

Kaja Heidenreich [1], Anne-Marie Slowther,[2] Frances Griffiths [2], Anders Bremer [3], Mia Svantesson[1]

[1]University Health Care Research Center, Faculty of Medicine and Health, Örebro University, Örebro, Sweden
[2]Warwick Medical School, University of Warwick, Coventry, UK
[3]Department of Health and Caring Sciences, Faculty of Health and Life Sciences, Linnaeus University, Växjö, Sweden

**Correspondence to**
Kaja Heidenreich;
kaja.heidenreich@regionorebrolan.se

## ABSTRACT

**Objective** The decision whether to initiate intensive care for the critically ill patient involves ethical questions regarding what is good and right for the patient. It is not clear how referring doctors negotiate these issues in practice. The aim of this study was to describe and understand consultants' experiences of the decision-making process around referral to intensive care.

**Design** Qualitative interviews were analysed according to a phenomenological hermeneutical method.

**Setting and participants** Consultant doctors (n=27) from departments regularly referring patients to intensive care in six UK hospitals.

**Results** In the precarious and uncertain situation of critical illness, trust in the decision-making process is needed and can be enhanced through the way in which the process unfolds. When there are no obvious right or wrong answers as to what ought to be done, how the decision is made and how the process unfolds is morally important. Through acknowledging the burdensome doubts in the process, contributing to an emerging, joint understanding of the patient's situation, and responding to mutual moral duties of the doctors involved, trust in the decision-making process can be enhanced and a shared moral responsibility between the stake holding doctors can be assumed.

**Conclusion** The findings highlight the importance of trust in the decision-making process and how the relationships between the stakeholding doctors are crucial to support their moral responsibility for the patient. Poor interpersonal relationships can damage trust and negatively impact decisions made on behalf of a critically ill patient. For this reason, active attempts must be made to foster good relationships between doctors. This is not only important to create a positive working environment, but a mechanism to improve patient outcomes.

## BACKGROUND

Admission to an intensive care unit (ICU) allows critically ill patients to access life-saving treatment. However, the short-term and long-term mortality of patients in the ICU is substantial.[1–3] In the UK, 20% of patients admitted to the ICU die in the unit and, additionally, 10% will die before hospital discharge.[4] The care received involves invasive, distressing and potentially harmful interventions, as

## Strengths and limitations of this study

► The phenomenological hermeneutic method enabled the meanings of the lived experiences of a morally complex decision-making process to be captured, resulting in a deeper understanding of this process.
► The sample included hospitals of different sizes and a range of clinical specialties, which provided a variety of experiences to inform the research question.
► The sample size was large for a phenomenological hermeneutical analysis, which challenged the analysis process and the depth of analysis.

well as psychological stress for both patients and their families.[5–8] Patients' experiences, reported after a stay in the ICU, have revealed strange and frightening memories as well as a process of struggling to find meaning in their ongoing lives.[9 10]

Given the burden of intensive care therapy and the limited prognosis, intensive care treatment will not be beneficial for all patients, and ward-based care or palliative care might be more beneficial.[11] The decision whether to refer and admit a patient to the ICU is often complex, time pressured and made in the context of clinical uncertainty with limited knowledge of the patient's wishes for their further care.[11 12] This complex situation was the departure for a multidimensional mixed methods project, Understanding and Improving the Decision-Making Process Surrounding Admission to the Intensive Care Unit, conducted at six NHS trusts in the Midlands, UK.[11–14 15] The present paper is a substudy of the wider project.

Previous research suggests that there is large variation in how decisions about admission to the ICU are made, depending on the individual clinician, unit, hospital and country characteristics.[16–19] The focus of previous research has been on justice and resource allocation[20–24] and factors that influence the decisions.[25 26] In a systematic

review, factors that correlated a decision were bed availability, severity of illness, referral ward or team, patient choice, do not resuscitate status, age and functional baseline.[27] Experiences of the decision-making *process* are less explored, but according to a recent systematic review of this literature, professional relationships, attitudes and communication between stakeholders as well as acknowledging uncertainty seemed important.[28] However, the described experiences have foremost been considered from the perspective of intensive care, and the perspective of referring clinicians is missing.[29 30] The Society of Critical Care Medicine offers guidelines for admission, discharge and triage, but only sparse guidance about the complex decision-making processes.[31]

The wider project used a focused ethnography method to describe and understand the decision-making process.[11 12] The project included the perspective of referring doctors from wards and emergency departments, as little is known about their experiences of the decision-making process. In addition, clinical experience is valuable in developing deeper knowledge about this complex decision-making process and, therefore, the perspective of consultant doctors was sought to develop further understanding of the process.

Thus, the aim of the present study was to describe and understand consultant doctors' experiences of the decision-making process around referral to intensive care.

## METHODS
### Design
This study had a qualitative design, using phenomenological hermeneutics as the analysis method.[32] The present paper is an analysis of a subset of the data collected in the ethnographic study in the National Institute for Health Research project.[11]

The phenomenological hermeneutical method was developed for the interpretation of lived experiences of ethically difficult situations, so-called 'lived ethics'.[32] Lived ethics, that is, how people act and think about real moral situations, are not easily accessible in empirical research because most people have profound difficulties explaining why their actions are morally good or bad. In this study, we have used the method to reveal the underlying ethics through interpreting the text through a dialectic movement between understanding and explanation.[32] The phenomenological element aims to describe and understand the lived experiences of 'actions, attitudes, relations or other human matters as ethically good or bad'.[32] The hermeneutical element draws on Ricœur's theory of interpretation and aims to move from what the text says to what meaning we can understand by performing a closer reading.[33]

### Participants and setting
Six UK hospitals were included in the larger project for diversity in type of hospital (university/non-university)

and size of ICU.[12] All consultants from departments likely to refer patients to the ICU were emailed an invitation to participate from the hospital's principal investigator. The goal was to recruit five consultants at each hospital. At hospitals with more than five volunteers, purposive sampling was used to achieve a spread of gender and clinical specialty. At two hospitals, where fewer than five volunteered, the principal investigators targeted consultants who were known to have opinions regarding intensive care referrals by sending a personal reminder. The inclusion process resulted in 27 consultants choosing to participate. The participants represented medical specialities (acute/emergency, general internal, respiratory, cardiology, renal, gastroenterology, haematology, endocrinology, infectious disease, neurology, geriatrics and oncology) and surgical specialties (general, colorectal, neurosurgery and orthopaedics). For demographic data, see table 1.

### Patient and public involvement (PPI)
The overall project had a lay advisory group and two PPI coinvestigators who contributed to the design, conduct and reporting of all aspects of the study. They did not contribute to the analysis of this data subset but were involved in the analysis of other elements of the larger project.

### Data collection
MS interviewed all but eight participants; the remaining were interviewed by A-MS or FG. Interviews were semistructured and conducted face-to-face. Participants were initially asked whether they had referred or considered referring any patients to intensive care during the previous 3 weeks and were encouraged to describe their experiences of making the decision about whether to contact the intensive care team. Probes explored reasons for contacting the ICU, the context of the process and its outcome. They were also asked about their general experience of the referral process and how it could be improved. Interviews were audio-recorded and transcribed verbatim; the average duration was 35 min (range 14–59 min) Interview guide, see online supplemental file 1.

| Table 1 Demographic information for the participants | |
|---|---|
| **Characteristics (n=27)** | |
| Gender, n (%) | |
| Male | 19 (70) |
| Female | 8 (30) |
| Age, mean (range) | 47 (31–59) |
| Years of experience, mean (range) | |
| In current specialty | 10 (0.2–22.0) |
| Since graduation | 22 (8–34) |
| Specialty, n (%) | |
| Medical specialties | 15 (56) |
| Surgical specialties | 7 (26) |
| Acute/emergency medicine | 5 (18) |

## Data analysis

The analysis included three steps: naïve reading, structural analysis and comprehensive understanding, which were intertwined in a dialectic movement.[32] All transcripts were read repeatedly and notes were made according to the whole. This initial interpretation of the whole was formulated into a naïve understanding, but iteratively revised during the analysis process. During the second step, the structural analysis, the data were decontextualised in order to explain the text through coding into meaning units which were grouped according to their meaning content. The meaning units were condensed, and themes and subthemes that shared similar meanings within the overall aim of the analysis were developed. Meaning units, subthemes and themes were continuously validated and refined by returning to the data as well as the naïve understanding. As the analysis unfolded, a deeper understanding of the meaning of the participants' experiences was captured through shifting between the descriptive and interpretative levels of the analysis. Due to the extensive amount of data, the software programme NVivo V.11 was used to facilitate the initial analysis process. During the writing up of the structural analysis, recategorisation continued in light of the whole dataset. In the final step of the analysis, a comprehensive understanding with a further interpretation of the text was developed using the theory of the 'Ethics of proximity' by Levinas.[34 35]

Throughout the process, analysis meetings were held between the coauthors to validate the findings. The SRQR reporting guideline was applied.[36]

## RESULTS

### Naïve reading

The initial interpretation of the text, iteratively revised, revealed that caring for a critically ill patient implies an obligation to urgently respond and take measures to secure a decision for the further care of the patient. The referring doctors experience the critically ill patient as being seriously affected by their illness, and the clinical situation is imprinted with the potential for death and suffering. They are aware of the consequences that their decisions will have on the patient's future life, and the moral responsibility of their decision is evident.

Making judgements about referral to the ICU is based on reflection about whether this escalation would be beneficial for the patient regarding outcome and future care trajectory. When there is certainty about the benefit of intensive care treatment, referrals appear to be straightforward for the clinicians and the referring process could have larger diversity. When they are convinced that palliative care or ward-based care is most appropriate for the patient, they do not ponder over referral to ICU at all.

However, they struggle with making decisions for patients who are experienced as being in a 'grey area', where the meaningfulness of further care is difficult to interpret and imprinted with uncertainty and doubts. This implies not being convinced whether escalation to intensive care would be good and right for the patient while at the same time not being confident with making a decision to provide ward-based care alone.

When the decision-making process is loaded with doubts, the way in which the process unfolds is important for trust in the final decision. A profound understanding of the critically ill patient's clinical situation, as well as the doubts of the referring doctor, emerges in the encounter between the doctors and the patient and/or their family.

### Structural analysis

The structural analysis revealed the three themes of 'burdensome doubt', 'emerging joint understanding' and 'responding to mutual moral duties' and further nine subthemes (table 2). An extended table 2 with more quotes could be found under online supplemental files 2.

#### Burdensome doubt

The consultants describe a struggle to make the right decision for the critically ill patient under difficult conditions while being highly aware of the risks of death and suffering. The burdensome doubt of the decision-making process signifies a need for trust in the process.

The doctors are affected by the gravity of the critical illness, which is imprinted with urgency and precariousness. The patients are perceived as being exhausted and often very affected by their critical condition. This places a moral responsibility on the referring doctor to make the right decision about the further care of the patient.

It was an extremely difficult decision … For me, admitting her to the intensive care unit and supporting her respiratory system was as much about the symptomatology and control and actually good palliative

| Table 2 | Structural analysis with themes and subthemes | | |
|---|---|---|---|
| **Themes** | **Burdensome doubt** | **Emerging joint understanding** | **Responding to mutual moral duties** |
| Subthemes | Being affected by the gravity of critical illness | Comprehension by seeing the patient at the bedside | Meeting in person to acknowledge concern |
| | Pondering about what is meaningful for the patient | Requiring senior experience to grasp clinical complexity | Protecting the patient through advocacy |
| | Ambivalence in searching for the patient's voice | Needing engagement for broadened perspectives | Building relationships for confidence |

care rather than … to turn this around. … that the patient's very conscious and gets very breathless, very worked up, very anxious and very frightened. (Haematology consultant, hospital 2)

Pondering about what is meaningful for the patient refers to considering whether further escalation to intensive care would be good and beneficial for this particular patient. This requires the doctor to obtain a clinical overview of the patient's medical situation in the face of an often rapid change in their condition. The clinical scenario is often experienced as being unclear with a lack of information and, by trying to articulate the precise clinical needs of the patient, they could ponder what intensive care could offer the patient.

I can do lots of technological things but I'm not sure it gives people a good death and I'm not sure it gives people a good quality of life for their last few weeks. (Internal medicine consultant, hospital 2)

Doctors experience ambivalence in searching for the patient's voice and involving the patient and family in the decision-making process. They acknowledge on the one hand that the patient and family are natural partners in the process, but, on the other hand, their voice is experienced by the doctors as ambiguous. The doctors reflect on the patient and family's ability to understand the different possibilities for the further care, particularly when the patients are often too sick and exhausted to voice anything at all. They reflect on how the patient and the family's wishes for treatment could be driven by fear as well as a lack of realism about what intensive care could achieve.

We should try and involve the patient, [but] she was clearly not in a position to be asked. We should try and ascertain her current and previous wishes. She was unable to give her current wishes … So we discussed with the family what they think she would have wanted and I think they were unsure as well. (Geriatric consultant, hospital 2)

### Emerging joint understanding

The decision-making process implies multiple perspectives and therefore needs to stimulate an emerging joint understanding of the patient's precarious situation from the perspectives of the different stakeholders. Understanding emerges through comprehending the clinical situation at the bedside, through the eyes of experienced clinicians as well as through engagement with broadened perspectives on the patient's situation. A profound understanding of the critical illness situation is essential for developing trust in decisions about the further care of the patient.

Comprehension by seeing the patient at the bedside means that the bedside encounter with the patient contextualises the medical information that is available from multiple other sources, such as files, tests and examinations, and, thereby, the unique situation of the particular patient can emerge. Seeing the patient at the bedside means gaining a comprehensive understanding of the patient's situation, and the importance of this in-person encounter with the patient extends to both the referring doctor and the intensivist.

First of all, undoubtedly the reality that it's a person, you see them, so if somebody phones me it's a list of numbers, blood pressure, observations I get because I'm a kidney physician. Whereas when you go to see somebody, they're real, they've got family and when you're then thinking about it you contextualise it, you think about this is a real person. (Nephrology consultant, hospital 3)

Patients could be in more substantial need of intensive care than the numerical physiological measures and telephone reports have suggested. Initial reluctance from the intensivist to admit a patient can rapidly change when they see the patient. When intensivists dismiss the need for admission to intensive care without seeing the patients, the referring doctors regard this as irresponsible practice.

Senior clinical experience at a consultant level is required to be able to grasp the clinical complexity and to understand critical illness as a dynamic condition. Being able to anticipate the unpredictability of the situation and to make anticipatory plans for further care of the patient helps to secure the safety of the patient as well as the doctor's own confidence in the unpredictable situation.

Especially when it's the patients that could get a lot worse, it's almost a seniority thing. They [junior doctors] don't know enough, haven't seen enough to realize that the information that you're imparting is because we're basing it on experience of things going worse. (Acute and emergency medicine consultant, hospital 1)

Needing engagement for broadened perspectives refers to the mutual responsibility of both the referring doctor and the intensivist to actively seek more information about the patient's situation in order to broaden their own perspectives and contribute to a more profound understanding of the patient's situation.

Referring doctors describe their need to obtain the perspective from intensive care on what can be done as well as a realistic view on what it is possible to do for a critically ill patient. They describe how they experience only being able to judge the most acute needs of the patient and acknowledge the clinical situation over a short period of time and that the intensivist's judgement contributes to a broader perspective by providing new insights and understanding. The intensivist is experienced as an important sounding board for the referring doctor.

You're not quite sure, you just know someone's ill and you're just trying to work out what is it that I need and then that consultant who I spoke to is very good

at sort of saying: 'Well what is it you're actually asking for? Let's work this out!' (Internal medicine consultant, hospital 1)

However, experiences of lack of engagement from the intensivist and a too-narrow perspective on the patient's situation mean a lack of understanding of the patient's difficult situation and an increased burden on the referral doctor with loss of trust.

That's a very easy thing to do, to walk in and say: 'Not fit for intensive care,' then wander back to your intensive care unit. That's quite an easy decision to make. You're not then the person who's there, dealing with that patient, that family saying: 'Why aren't you treating my mum?' (Colorectal surgeon, hospital 5)

### Responding to mutual moral duties

Fulfilling moral duties is part of the process of gaining trust in the decision-making process, and this is achieved in the encounter between the doctors.

Meeting in person to acknowledge concern refers to the need to meet face-to-face with the intensivist. Meeting in person facilitates discussion; new questions are answered directly; and it is easier to make further plans for the patient's care. The referring doctor's concern for the critically ill patient can be acknowledged, and more confidence in the decision can be gained by sharing the burden of making such a complex decision.

People who come to see the patient, make a proper assessment, talk to me and explain to me, I can ask questions and we come to an agreement. I think that's straightforward. There the trust grows very quickly. (Respiratory physician, hospital 6)

The referring doctors see themselves as having a duty to advocate on behalf of the patient when there is disagreement and lack of confidence about the further care of the patient. This includes arguing for the possible benefit of intensive care. Advocacy is seen as being important when bed capacity is a concern. The referring doctors are aware of the intensivists' concern of capacity. However, they see that their obligation is to strictly focus on the needs of their critically ill patient:

Beds are tough but, that's not a part of the decision-making; is the patient going to get better care on ICU than where they are now? Can we do something to intervene and then if the decision-making is 'yes' and if appropriate for the patient then the final [a bed] is secondary. (Haematology consultant, hospital 6)

Advocacy also involves arguing for the time needed in ICU to get a clearer picture of the medical condition. Admission with a restriction in length of stay or type of intervention could be offered to give the patient a chance, but at the same time restricts long-term suffering. These negotiations are a way of operationalising the duty of protection from harm.

Building relationships between the referring doctor and the intensivist facilitates confidence and further trust in the decision. This is achieved through learning to know each other while sharing the care for patients. Although the intensivist makes the formal decision about an admission, the referring doctors describe their own duties in making discerning and adequate clinical judgements to contribute to a necessary and mutual confidence in the process.

A strong, positive relationship could mean that the referring doctors ignore the formal referral system and directly contact senior intensive care colleagues to discuss their patients. The referring doctors also describe having a personal and extended responsibility for patients being cared for on the ICU, particularly for patients whom they have known for a long time, but also patients suffering from complicated or rare diseases, where they regard their knowledge and experience to be important. This responsibility is grounded in a personal relationship of mutual confidence and support that the referring doctors are eager to foster.

I have a good relationship with the more senior ICU consultants and I suppose it all is dependent on trust. I trust their judgement and they trust my judgement and that is based on years of working with each other and seeing what we do and how we make decisions … it comes from experience, so if I referred patients to the intensive care unit and they took them and then felt time and time again they weren't appropriate referrals they would stop trusting my judgement. (Nephrology consultant, hospital 3)

### Comprehensive understanding

The structural analysis of referring doctors' experiences of the decision-making process revealed the burdensome doubt of the referring doctor, how to contribute to an emerging, joint understanding of the patient's situation and mutual moral duties that need response. Previous research has shown tension and misunderstandings and the need for improved communication between ICU and ward teams.[11 13 28] Our study shows from the perspective of referring consultants, how trust in the decision-making process could be enhanced through the way the process unfolds in the relations between the patient and the stake holding doctors. When the clinical scenario does not have an obvious right or wrong answer to what ought to be done, *the way* in which the decision is made and *how* the process unfolds are important for gaining trust in the decision-making process. The further comprehensive understanding of the data was developed in light of the writings of Levinas and the concept of ethics of proximity.[34 35]

The face-to-face encounter, as well as the moral responsibility that manifests itself in this encounter, is central to Levinas' philosophy.[37] It deals with the existential character of human relations and signifies a search for deeper patterns of being human in relation

to others. Levinas describes how, in the encounter with 'The Other's Face', physical presence is not foremost; rather, we are called by The Other's Face and responding to that 'otherness'. The Other is substantially different from me, and this otherness establishes a basic asymmetry in the relation between us. Based on the asymmetrical relation of otherness as well as proximity when encountering each other's faces comes the moral claim of protection and responsibility.

Seeing the critically ill patient in person bedside is important in the decision-making process and is true for both the referring doctor as well as the intensivist. This Gestalt assessment of the patient has previously been shown to be important in the decision-making process.[14] Seeing the patient could be further interpreted, in light of Levinas' philosophy, as being an expression of the demand to encounter the critically ill patient as The Other's Face, and so recognise, through a clinical proximity, the moral claim for responsibility in a critical and vulnerable situation. Seeing the face of the patient adds substantial moral information to the process—what is at stake and what ought to be done—over and above the medical information. In the asymmetrical relation with the other, we encounter, according to Levinas, the other's, as well as our own, vulnerability, and the empirical findings reveal what could be interpreted as a substantial vulnerability.[35] The critically ill patient is perceived as being vulnerable, imprinting gravity in the decision-making process. The referring doctor's burdensome doubt exposes their own vulnerability—being close to the existential dimensions of life and burdened by the seriousness and uncertainty of the decision.

Levinas describes how, in the encounter with The Other's Face, we are called to protect and take responsibility. This is in line with the referring doctors' advocacy for the patient when they think the patient should be admitted to the ICU and could be seen as a clinical outlook of protection and responsibility. However, the moral claim of responsibility and protection will not in itself inform which decision to make or what is the right thing to do, but could be seen as an underlying prerequisite for making a moral judgement, through clinical and moral proximity, with the patient as well as with the intensivist. When the proximity in experienced as responsible by the referring doctors as highlighted in the structural analysis, trust in the decision-making process could be enhanced.

Proximity in ethics has been described as an important part of moral perception. Moral perception is used to describe our capacity to acknowledge morally salient aspects of a particular situation.[38] Moral perception, to see that something matters morally, can be distinguished from moral reasoning, a more conscious deliberation about action and reasons for actions.[39] In the present study, the importance of clinical proximity could also be interpreted as being important for the moral perception of a critical situation. Being close to the patient, we found that the moral obligation of the doctors to always see the patient at the bedside, could contribute to identifying morally salient aspects of the patient's situation that are not detectable over the phone or in notes and could be seen as a component of enhancing trust in the decision-making process.

The findings contribute to an increased understanding of the ethical as well as clinical basis for the known importance of relationships between clinicians and their patients in the complex decision-making process of referral to intensive care. This model of the interaction between the intensive care and referring clinical teams is characterised by doubt, resolved through discussion to a shared understanding and assuming of moral responsibility, and from that trust in the decision-making process could emerge.

The moral understanding that this analysis provides supplements the clinical understanding by identifying important duties, trusts and relationships. When these duties and relationships break down, it is likely to negatively impact the patients who rely on the care that can only be provided as part of a broad hospital team as well as negatively impact working relationships.

## METHODOLOGICAL CONSIDERATIONS

The data were not collected with the intention of using a phenomenological hermeneutic analysis method, which could pose a potential threat to validity. However, we regarded the data as being rich with the possibility of capturing meanings of lived experiences. The first author, who led the analysis, did not conduct any interviews, which could be seen as a weakness. However, the data analysis process was conducted under constant critical review and reflection by the coauthors with different kinds of preunderstandings. The first author (KH) has clinical experience as a referring medical doctor and the rest have clinical experience from general practice (A-MS and FG), intensive care nursing (MS) and ambulance care (AB). Three were involved in the data collection and the rest of the project (MS, A-MS and FG) and coming from outside (AB), ensuring trustworthiness.

It can be argued that the sample size was too large for this method, thus challenging the depth of the analysis when dealing with the large quantity of data. However, it may also be regarded as a strength, as we captured varied narratives from different clinical specialties as well as types of hospital.

The clinical context of the study is the UK, but the findings might be transferable to similar organisations of healthcare in developed countries. However, this is in the eye of the reader to judge.

## CONCLUSION

This study has provided an increased understanding of the interaction between the moral and clinical aspects around referral to intensive care. The paper has

articulated a rationale for the important joint assessment of a patient at the bedside that is well recognised but often neglected.

The findings highlight the importance of trust in the decision-making process and how the relationships between the stakeholding doctors are crucial to support their moral responsibility to the patient. Poor interpersonal relationships can damage trust and negatively impact decisions made on behalf of a critically ill patient. For this reason, active attempts must be made to foster good relationships between doctors. This is important to create not only a positive working environment but also a mechanism to improve patient outcomes.

The study has clinical implications for guidance to support the process of referral to intensive care and should be used to improve the referral process. Hospitals should consider specifying best practice in referrals, including joint bedside assessment, as described in the wider ICU referral and admission decision-making project.[12 11]

**Acknowledgements**  We thank the consultants who participated in the study. Dr Chris Bassford was chief investigator for the larger project looking at decision making around admission to intensive care in which the data analysed in this paper were collected. Dr Bassford has provided helpful comments on the revised manuscript to make it more readable for clinicians and suggested revisions to the conclusion to highlight the clinical implications.

**Contributors**  KH: conducted the coding, led the analysis and wrote the manuscript. A-MS: conducted two interviews, scrutinised the analysis and reviewed the manuscript. FG: led the project work package in which data were collected, conducted six interviews and reviewed the manuscript. AB: scrutinised the analysis and reviewed the manuscript. MS: conducted 19 interviews, scrutinised the analysis and reviewed the manuscript.

**Funding**  This article presents independent research funded by the National Institute for Health Research under the Health Services and Delivery Research programme (project number 13/10/14).

**Patient consent for publication**  Not required.

**Ethics approval**  The project obtained ethics approval from the NHS Health Research Authority and the West Midlands-Coventry & Warwickshire research ethics committee (15/WM/0025, IRAS project ID 160660). Participants gave written consent prior to the interview.

**Data availability statement**  No data are available. No additional data available.

**ORCID iDs**
Kaja Heidenreich http://orcid.org/0000-0002-1983-9813
Frances Griffiths http://orcid.org/0000-0002-4173-1438
Anders Bremer http://orcid.org/0000-0001-7865-3480

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
