## [Reviewer comments · BMJ Open]

ARTICLE DETAILS

TITLE (PROVISIONAL)	UK Consultants experiences of the decision-making process around referral to intensive care: an interview study
AUTHORS	Heidenreich, Kaja; Slowther, Anne-Marie; Griffiths, Frances; Bremer, Anders; Svantesson, Mia

VERSION 1 – REVIEW

REVIEWER	Christina L. Cifra University of Iowa Carver College of Medicine, USA
REVIEW RETURNED	09-Oct-2020

GENERAL COMMENTS	The authors describe a qualitative study consisting of semi-structured interviews of consultant doctors who regularly refer patients to the ICU in 6 UK hospitals. The objective of the study is to illuminate the lived experiences of consultants in the decision-making process when referring patients to intensive care. This is a well written and rigorous qualitative study that explores an understudied topic of the experiences of referring clinicians who transfer patients to the ICU. I am impressed with the rigor of the work and also with the care and thought that went into writing the manuscript. It is highly readable, the format is easy to follow and very understandable even for readers who may be less familiar with qualitative methods. The following are some comments on the work including points for potential improvement. Abstract: • Although very readable, the authors tend to lapse into jargon at times. The objective, for example, can be stated more simply. Instead of “to illuminate the meaning..,” may it be revised to “to describe and understand...”?• In Results, it is unclear whose trust is being referred to – trust between referring and ICU clinicians? Between referring clinicians and patients? Also, the statement, “can be nurtured through the way the process unfolds” is confusing and was much better expressed a couple of sentences later, “how the process unfolds...”• The Conclusion as stated can be revised into simpler language. Again, trust between whom? What is “responsible proximity”? Background • Very well written and readable. Set up is good and clearly identified gaps in knowledge which led to this study.• Good that authors immediately placed this study within the larger (former) study’s perspective.
---

	 • Did the methods include participant observation or just interviews? Methods  • Good description of qualitative methodology with a neat rundown of the chronological steps in the process and also the iterative analysis. • Although I appreciate the background explanation of the phenomenological hermeneutical method, I suggest editing this section down to the minimum explanation needed by the reader to understand the method employed so as not to distract from the rest of the work. • Good description of selection of study participants. Is it possible to compare the characteristics of the study cohort to the general population of consultant physicians? For example, is the proportion of 30% women physicians representative of the general cohort? • Under Ethical considerations, suggest to delete information re: lay advisory group and contributions of research team members since there is a separate section at the end for this information. • Were there participants with no recent ICU transfer experience or did all have a transfer in the recent 3 weeks before their interview? Results  • I appreciate the Table breaking down findings of themes and sub-themes. • Although I appreciated the clear descriptions of each theme plus the section under Comprehensive understanding, I suggest that the authors edit this section down to perhaps half of the original content. Edit out repetitive explanations and descriptions/analyses so as readers can remain engaged in the material. • It is only under the authors' discussion of the theme, emerging joint understanding that it dawned on this reader that they were discussing the relationship between referring and ICU clinicians. This should be made clear earlier on in the manuscript. • Good examples of representative quotes from subjects. Conclusion  • See prior comment re: conclusion in Abstract. • Paragraph about COVID-19 seems tacked on and came out of the blue. Would delete this last paragraph.
--	---

REVIEWER	Professor Stephen Brett Imperial College London
REVIEW RETURNED	17-Oct-2020

GENERAL COMMENTS	Personally I found this an interesting and thought provoking paper, but I do have some concerns and suggestions - some of this is about who the investigators anticipate reading the paper. As written, the social science aspects, technical description and theoretical exploration are obscuring the clinical message and its importance. I suspect few intensivists or ward based consultants- other than people with an interest such as myself- would get to the end of this, which would be a shame. If the authors (and editors) are content for the paper to be read only by academics in the decision making/qualitative field that is different. I would suggest the authors review the presentation of their paper with a suitable intensive care or medical consultant who may be able to re emphasise the clinically important elements and perhaps alter the balance of data presented and discussion (Dr Chris Bassford in
---

	Warwick/Coventry is very local and has a major track record in decision making research in intensive care). Non-essential jargon needs ruthlessly removing; e.g. few, if any, clinical readers will progress beyond the word "epistemology". I am comfortable with the merging of results and discussion in the way the authors have- this helps the story. The relationship to and a brief description of, the overall programme needs explaining In terms of specific things at this stage.  1. The words "moral" and "proximity" are used extensively and in ways that may not be familiar to many readers. I think these concepts need defining clearly for a broader readership- used as they are is confusing. 2. I would suggest de-agregating Table 1 describing the participants individually (speciality, some measure of experience, DGH vs. Teaching, gender etc.)- if that would be allowed, also the backgrounds of those undertaking the interviews in a bit more detail. 3. I couldn't find a copy of the interview schedule/topic guide- this should be somewhere. 4. There needs to be far greater presentation of the data for this to be persuasive. What I have found helpful, and often been asked to provide, is a table with the themes identified - along with quotes which generated the themes. This could be a supplement- but it does help with the validity of the discussion and conclusions. I couldn't find such a table.It would be an explosion of Table 2. The quotes presented in the text are very helpful but I think there needs to be more. 5. p11 line 38 et seq. is a fairly deep discussion of theory- which will not resonate with a clinical readership and I think would be better summarised briefly and suitably referenced. If the authors feel this is essential- then I am not sure BMJOpen is the most appropriate journal- but that would be for the editors to give an opinion. 6. There is a literature on decision making around ICU admission decisions and it would be worth briefly summarising what this is and how the current study progresses things. The current conclusion could be then be sharpened up.
--	---

REVIEWER	Frank Kiwanuka University of Eastern Finland, Finland
REVIEW RETURNED	11-Nov-2020

GENERAL COMMENTS	Thank you for the opportunity to review this manuscript. The manuscript substantially provides interesting findings on the decision making process in the ICU context. However, a few comments need to be addressed.  1. Abstract The abstract clearly summarizes the study. 2. Introduction and background: the section is well written. Used good sources to highlight literature on the topic. However, the phenomenon the problem would have been discussed further in the background. 3. Methods: the approaches are adequately described and are appropriate for this study. The henomenological hermeneutic approach is congruent with the philosophical perspective that the authors investigated. However, a statement depicting the author(s) cultural or theoretical location to the phenomenon being investigated needs to be probably added. This was a sub-study of a larger study; however, the readers and other researchers would benefit from understanding whether there
---

	was influence of the researcher on the research, and vice-versa? and if any, how was this addressed? 4. Results: the results are well organized to answer the research questions. The author(s) substantially presented the participants, and their voices. 5. Discussion: The discussion is well written ho can be elaborated with more interpretations aligned with study’s new knowledge in the context of what is already known.
--	---

VERSION 1 – AUTHOR RESPONSE

Reviewers’ comments	Authors’ responses
Reviewer 1 1. This is a well written and rigorous qualitative study that explores an understudied topic of the experiences of referring clinicians who transfer patients to the ICU. I am impressed with the rigor of the work and also with the care and thought that went into writing the manuscript. It is highly readable, the format is easy to follow and very understandable even for readers who may be less familiar with qualitative methods.	Thank you! We have carefully read all the comments on the manuscript and have revised it according to your suggestions.
2. Abstract: the authors tend to lapse into jargon at times. The objective, for example, can be stated more simply. Instead of “to illuminate the meaning,” may it be revised to “to describe and understand...”? In Results, it is unclear whose trust is being referred to – trust between referring and ICU clinicians? Between referring clinicians and patients? Also, the statement, “can be nurtured through the way the process unfolds” is confusing and was much better expressed a couple of sentences later, “how the process unfolds” The Conclusion as stated can be revised into simpler language. Again, trust between whom? What is “responsible proximity”?	We have carefully revised the language to avoid unnecessary wording and to make it more readable. Thank you, we see that this has been unclear and have revised the language. The studied phenomenon is the decision-making process and the result regards the consultants’ trust in this process. The text has been revised to hopefully be more clear.

	We agree that the term “responsible proximity” might be difficult to understand and have revised the language to increase clarity. We have also revised the language in the conclusion to be easier to understand.
3. Background  • Very well written and readable. Set up is good and clearly identified gaps in knowledge which led to this study. • Good that authors immediately placed this study within the larger (former) study’s perspective. • Did the methods include participant observation or just interviews? 	Thank you. We have revised the section about the main study as we see that it could be misunderstood which type of data the manuscript describes.
4. Results  • I appreciate the Table breaking down findings of themes and sub-themes. • Although I appreciated the clear descriptions of each theme plus the section under Comprehensive understanding, I suggest that the authors edit this section down to perhaps half of the original content. Edit out repetitive explanations and descriptions/analyses so as readers can remain engaged in the material. 	The result section has been shortened and revised aiming to avoid unnecessary repetitions and to become more readable. While, we have tried to make the structural analysis more accessible for clinicians and easier to read, we consider the theoretical approach of the comprehensive understanding as important for understanding of the studied phenomenon. We have, thoughtfully revised the language to increase readability and the content of the comprehensive understanding is reduced to half of the original.
5. • It is only under the authors’ discussion of the theme, emerging joint understanding that it dawned on this reader that they were discussing the relationship between referring and ICU clinicians. This should be made clear earlier on in the manuscript.  • Good examples of representative quotes from subjects. 	We agree. We have tried to clarify this under each theme.

6. Conclusion  • See prior comment re: conclusion in Abstract. • Paragraph about COVID-19 seems tacked on and came out of the blue. Would delete this last paragraph. 	We have revised this according to your suggestions.
Reviewer 2	
1. Personally I found this an interesting and thought provoking paper, but I do have some concerns and suggestions - some of this is about who the investigators anticipate reading the paper. As written, the social science aspects, technical description and theoretical exploration are obscuring the clinical message and its importance. I suspect few intensivists or ward based consultants- other than people with an interest such as myself- would get to the end of this, which would be a shame. If the authors (and editors) are content for the paper to be read only by academics in the decision making/qualitative field that is different. I would suggest the authors review the presentation of their paper with a suitable intensive care or medical consultant who may be able to reemphasise the clinically important elements and perhaps alter the balance of data presented and discussion (Dr Chris Bassford in Warwick/Coventry is very local and has a major track record in decision making research in intensive care). Non-essential jargon needs ruthlessly removing; e.g. few, if any, clinical readers will progress beyond the word "epistemology".	Thank you for thorough reading and valuable comments on how to present the results in a more approachable way for clinicians. We have rewritten and shortened the section "Structural analysis" with special emphasis on making this part of the results easier for clinicians to read and understand. We have also revised the result section "Comprehensive understanding" with the same aim and the content is reduced to half of the original. However, we do think that the theoretical perspectives contribute to a further understanding of this complex issues which could contribute to important insights and implications for practice. The manuscript is a part of a larger project where other publications might be experienced as easier for clinicians to read. Empirical ethics is an increasingly recognised approach to exploring complex decision-making and relationships in clinical practice which aims to add important insights into the lived reality of health care professionals and patients. By drawing on both the empirical data and philosophical theory we hope to explore and explain the moral dimension of the decision-making process. We have edited our manuscript to remove jargon and reduce discipline specific language, and to emphasise further the clinical relevance

	of our analysis. We hope this will help clinician readers. Chris Bassford was the CI on the larger project looking at decision making around admission to intensive care that this study is part of. Chris has now provided helpful comments on the manuscript to make it further readable for clinicians as well as revision of the conclusions and clinical implications.
2. I am comfortable with the merging of results and discussion in the way the authors have- this helps the story. The relationship to and a brief description of, the overall programme needs explaining In terms of specific things at this stage.	The manuscript is a part of a larger intensive care project. We have tried to clarify this sub study's part in the larger programme.
3.The words "moral" and "proximity" are used extensively and in ways that may not be familiar to many readers. I think these concepts need defining clearly for a broader readership- used as they are confusing.	We have revised the results to increase clarity for the reader and avoid unnecessary jargon.
4. I would suggest de-aggregating Table 1 describing the participants individually (speciality, some measure of experience, DGH vs. Teaching, gender etc.)- if that would be allowed, also the backgrounds of those undertaking the interviews in a bit more detail.	We have added more detailed information about the speciality of the participants and a reference for details about the size of the hospitals. Gender and measure of experience are displayed in the Table 1. We have displayed more information about the authors that conducted the interviews in the section "Methodological considerations" and under the section "Author statements" in the end of the manuscript.
5. I couldn't find a copy of the interview schedule/topic guide- this should be somewhere.	The interview guide has now been added as a supplementary file.

6. There needs to be far greater presentation of the data for this to be persuasive. What I have found helpful, and often been asked to provide, is a table with the themes identified - along with quotes which generated the themes. This could be a supplement- but it does help with the validity of the discussion and conclusions. I couldn't find such a table. It would be an explosion of Table 2. The quotes presented in the text are very helpful but I think there needs to be more.	Thank you, good suggestion. We have extended Table 2 and added quotes, now displayed as a supplementary file.
7. p11 line 38 et seq. is a fairly deep discussion of theory- which will not resonate with a clinical readership and I think would be better summarised briefly and suitably referenced. If the authors feel this is essential- then I am not sure BMJOpen is the most appropriate journal- but that would be for the editors to give an opinion.	The comprehensive understanding is in the phenomenological hermeneutical method regarded as a part of the results. The results from the structural analysis is further interpreted in light of relevant theory in the Comprehensive understanding. Even if the text might be difficult to grasp for clinicians and ethical theory might be unfamiliar to them, these perspectives could give substantial insights into a complicated clinical issue. We think that theoretical knowledge with relevance for medical practice actually is most suitable presented in a setting of empirical data. However, we have carefully revised the text to avoid unnecessary wording and jargon which might obscure the clinical message. The length of the text is shortened and is now half of the original. We actually think BMJ Open's general approach to medicine is appropriate for this article and have noticed that the journal has published empirical work with this method previously.
8. There is a literature on decision making around ICU admission decisions and it would be worth briefly summarising what this is and how	We have revised the background to further present previous knowledge of the decision-making process and we have added new references.

the current study progresses things. The current conclusion could be then be sharpened up.	The main project is now published as an article and this reference is added as well as an ethical analysis of the ethical conflicts in the decision-making process. We agree that the conclusion was too vague and have tried to make it more precise.
Reviewer 3	Thank you for positive comments. The background is revised to further highlight previous research about the decision-making process. We have also tried to further highlight that the studied phenomenon is the decision-making process of referral to intensive care.
1. Thank you for the opportunity to review this manuscript. The manuscript substantially provides interesting findings on the decision making process in the ICU context. However, a few comments need to be addressed. The abstract clearly summarizes the study. Introduction and background: the section is well written. Used good sources to highlight literature on the topic. However, the phenomenon the problem would have been discussed further in the background.	
2. Methods: the approaches are adequately described and are appropriate for this study. The phenomenological hermeneutic approach is congruent with the philosophical perspective that the authors investigated. However, a statement depicting the author(s) cultural or theoretical location to the phenomenon being investigated needs to be probably added. This was a sub-study of a larger study; however, the readers and other researchers would benefit from understanding whether there was influence of the researcher on the research, and vice-versa? and if any, how was this addressed?	We have added information about the clinical and cultural background of the authors under the section "Methodological considerations" and professional background in the author statement.

	We have added more information about the authors in the Author's statement and under the section "Methodological consideration" to hopefully clarify how the authors in the present paper were engaged in the larger project.
3. Results: the results are well organized to answer the research questions. The author(s) substantially presented the participants, and their voices. Discussion: The discussion is well written who can be elaborated with more interpretations aligned with study's new knowledge in the context of what is already known.	Thank you. We have added a new systematic review on previous research in the background and we align our results with this in the comprehensive understanding

VERSION 2 – REVIEW

REVIEWER	Professor Stephen Brett Imperial College London, UK
REVIEW RETURNED	27-Feb-2021
GENERAL COMMENTS	There is still some technical language and perhaps the authors feel that to remove this would undermine the credibility of the paper in a social science context. However, the vast majority of my concerns have been addressed and I enjoyed reading the revised MS- which I believe to be an important contribution. Stephen Brett Imperial College London